

# Isotopic biomonitoring of anthropic carbon emissions in a megalopolis

Edison A. Díaz-Álvarez[1] and Erick de la Barrera[2]

[1] Instituto de Investigaciones Forestales, Universidad Veracruzana, Xalapa, Veracruz, Mexico
[2] Instituto de Investigaciones en Ecosistemas y Sustentabilidad, Universidad Nacional Autónoma de México, Morelia, Michoacán, Mexico

## ABSTRACT

Atmospheric pollution has become a serious threat for human health and the environment. However, the deployment, operation and maintenance of monitoring networks can represent a high cost for local governments. In certain locations, the use of naturally occurring plants for monitoring pollution can be a useful supplement of existing monitoring networks, and even provide information when other types of monitoring are lacking. In this work, we (i) determined the tissue carbon content and the $\delta^{13}C$ values for the epiphytic CAM bromeliad *Tillandsia recurvata* and the relationship of both parameters with the existing CO concentrations in the Valley of Mexico basin and (ii) mapped the spatial distribution of such elemental and isotopic composition for this plant within the basin, in order to assess its potential as an atmospheric biomonitor of carbon monoxide, a pollutant with important repercussions on public health. The CO concentrations in the basin ranged from 0.41 ppm at rural locations to 0.81 ppm at urban sites. The carbon content of *T. recurvata*, which averaged 42.9 ± 0.34% (dry weight), was not influenced by the surrounding CO concentration. In contrast, the $\delta^{13}C$ depended on the sites where the plants were collected. For example, the values were −13.2‰ in rural areas and as low as −17.5‰ in an urban site. Indeed, the isotopic values had a positive linear relationship with the atmospheric CO concentrations. Given the close relationship observed between the isotopic composition of *T. recurvata* with the CO concentrations in the Valley of Mexico, the $\delta^{13}C$ values can be useful for the detection of atmospheric carbonaceous emissions.

## INTRODUCTION

Atmospheric pollution has become a serious threat for human health and the environment. This is especially worrying for populous cities, which is a frequent case throughout Latin America (*Kampa & Castanas, 2008*; *Rioja-Rodríguez et al., 2016*). For example, Mexico City, with vigorous industrial and household activities, as well as numerous motor vehicles of all classes, has seen an increase of emissions of different pollutants to the atmosphere. This has resulted in a higher incidence of respiratory and cardiovascular diseases, which already cause at least 9,600 premature deaths annually just

Corresponding author
Erick de la Barrera,
delabarrera@unam.mx

in this megalopolis and 20,000 in the whole country (*Stevens et al., 2008*; *Instituto Nacional de Estadística y Geografía (INEGI), 2011*). In addition, atmospheric pollution is one of the leading causes of biodiversity loss and ecosystem change both, from nitrogen deposition and the release of greenhouse gas emissions (*Sala et al., 2000*; *Rockström et al., 2009*; *Hooper et al., 2012*).

Carbon monoxide (CO) is among the main atmospheric pollutants with public health repercussions. Not only can acute exposure to high concentrations of CO lead to death, but the chronic exposure to low concentrations of this gas has been associated with cardiovascular and neurological damage (*World Health Organization WHO, 1999*; *Townsend & Maynard, 2002*; *Prockop, 2005*; *Chen et al., 2007*). CO results from the combustion of carbonaceous fuels, which also produces carbon dioxide. The proportion in which they are emitted depends on the quality of the combustion. For example, if a motor works in optimal conditions, that is, when the mixture of air, fuel and temperature inside an automobile engine is ideal, a complete combustion of the fuel is achieved, resulting in a complete oxidation of carbon and the subsequent emission of $CO_2$, predominantly. For example, 3.8 L of gasoline, whose weight is 2.7 kg, can emit 9 kg of $CO_2$ and ideally very low or insignificant amounts of CO (*Salameh, 2014*). Nevertheless, when the engine conditions are not optimal, an incomplete combustion/oxidation generates higher emissions of CO; this generally occurs in the engines of old cars and during heavy traffic congestions, which are frequent in the Valley of Mexico, where the vehicular fleet commonly exceeds 20 years of operation (*Williams, 1990*; *Turnbull et al., 2011a*; *Silva, Arellano & Worden, 2013*; *Salameh, 2014*; *SEDEMA Secretaría del Medio Ambiente de la Ciudad de México, 2016a*).

A characterization of air quality, by means of monitoring the concentration and distribution of atmospheric pollution, is thus imperative, including the monitoring of carbon emissions. However, the deployment, operation and maintenance of monitoring networks can represent high costs that in some cases exceed the budget and priorities of the local governments (*Díaz-Álvarez et al., 2019*). For example, only 77 cities from 17 countries in Latin America and the Caribbean have public information about atmospheric pollution, while 16 countries in the region do not release data at all (*Rioja-Rodríguez et al., 2016*). In Mexico, federal environmental and health regulations mandate the deployment of air quality monitoring networks for cities whose population exceeds half a million and for settlements with emissions surpassing 20,000 tons of regulated pollutants per year (*SEMARNAT, 2012*). However, as it occurs all over Latin America and the Caribbean, the monitoring of air quality is not evaluated in many localities as required (*Rioja-Rodríguez et al., 2016*; *Instituto Nacional de Ecología y Cambio Climático (INECC), 2017*).

The use of naturally occurring plants as biomonitors can supplement existing air quality monitoring systems and in some cases, when a monitoring system is lacking, be utilized for the early detection of increasing atmospheric pollution, considering that the elemental and isotopic composition of plant tissues can respond to the concentration of some pollutants (*Díaz-Álvarez, Lindig-Cisneros & de la Barrera, 2018*; *Díaz-Álvarez et al., 2020*). Once the origin and concentration of the pollution is determined, reduction

and mitigation actions can be implemented. A particularly suitable group of plants for biomonitoring are those that depend exclusively from atmospheric sources for their mineral nutrition, commonly named "atmospheric plants" (*Markert, Breure & Zechmeister, 2003*; *Vianna et al., 2011*; *Pellegrini et al., 2014*; *Díaz-Álvarez & de la Barrera, 2018*; *Díaz-Álvarez, Lindig-Cisneros & de la Barrera, 2018*). One of such species is *T. recurvata* (L.) L (*Schmitt, Martin & Luttge, 1989*). This CAM bromeliad is distributed from the southern United States to Argentina and Chile, and it is commonly found growing on different built structures in cities (*Schrimpff, 1984*; *Díaz-Álvarez & de la Barrera, 2018*). Additionally, the plant can remain physiologically active year-round and, thanks to its absorptive trichomes, carry out bioaccumulation of different atmospheric pollutants including heavy metals, polycyclic aromatic hydrocarbons, nitrogen, sulfur and carbon (*Schrimpff, 1984*; *Zambrano et al., 2009*; *Castañeda Miranda et al., 2016*; *Díaz-Álvarez & de la Barrera, 2018*: *Piazzetta, Ramsdorf & Maranho, 2018*). Although, the carbon isotopic composition of this plant has been reported for polluted and non-polluted sites, this is the first time that a specific relationship between carbon emissions and the plant's responses is determined.

By means of an extensive sampling throughout the Valley of Mexico, we (i) determined the carbon content and the $\delta^{13}C$ values for the epiphytic CAM bromeliad *T. recurvata* and their relationship with the prevailing CO concentrations, and (ii) mapped the spatial distribution of such elemental and isotopic composition of this plant in the basin, in order to assess the potential that this plant has as an atmospheric biomonitor of CO.

# MATERIALS AND METHODS

## Study region

The study was conducted in the Valley of Mexico basin which covers an area of 7,500 km$^2$, with a mean elevation of 2,240 m, and a mean annual precipitation of 600 mm that can reach up to 1,300 mm in the surrounding mountains at 5,400 m (Fig. 1; *Calderón & Rzedowski, 2005*; *Servicio Meteorológico Nacional (SMN), 2016*). The basin includes portions of the states of Hidalgo, Mexico, and Mexico City. Pachuca, the capital of Hidalgo, at the north has a population of 3 million. At the southern portion of the Valley sits Mexico City, whose population reaches 20 million. Additionally, various small towns and settlements with industrial or agricultural activities contribute to the 30 million inhabitants of the basin (*Instituto Nacional de Estadística y Geografía (INEGI), 2011*; *Díaz-Álvarez & de la Barrera, 2018*).

## Atmospheric CO concentration and biomonitoring

The Mexico City environmental authority has deployed an Automatic Atmospheric Monitoring Network (http://www.aire.cdmx.gob.mx/default.php) comprised of 33 stations, which monitor different parameters, including wet N deposition, $O_3$, NOx, $NO_2$, NO, $PM_{10}$ $PM_{2.5}$ and CO (Fig. 1). We calculated the mean concentration of CO (ppm) between January and November 2014 for each one of the 21 stations that recorded this parameter.

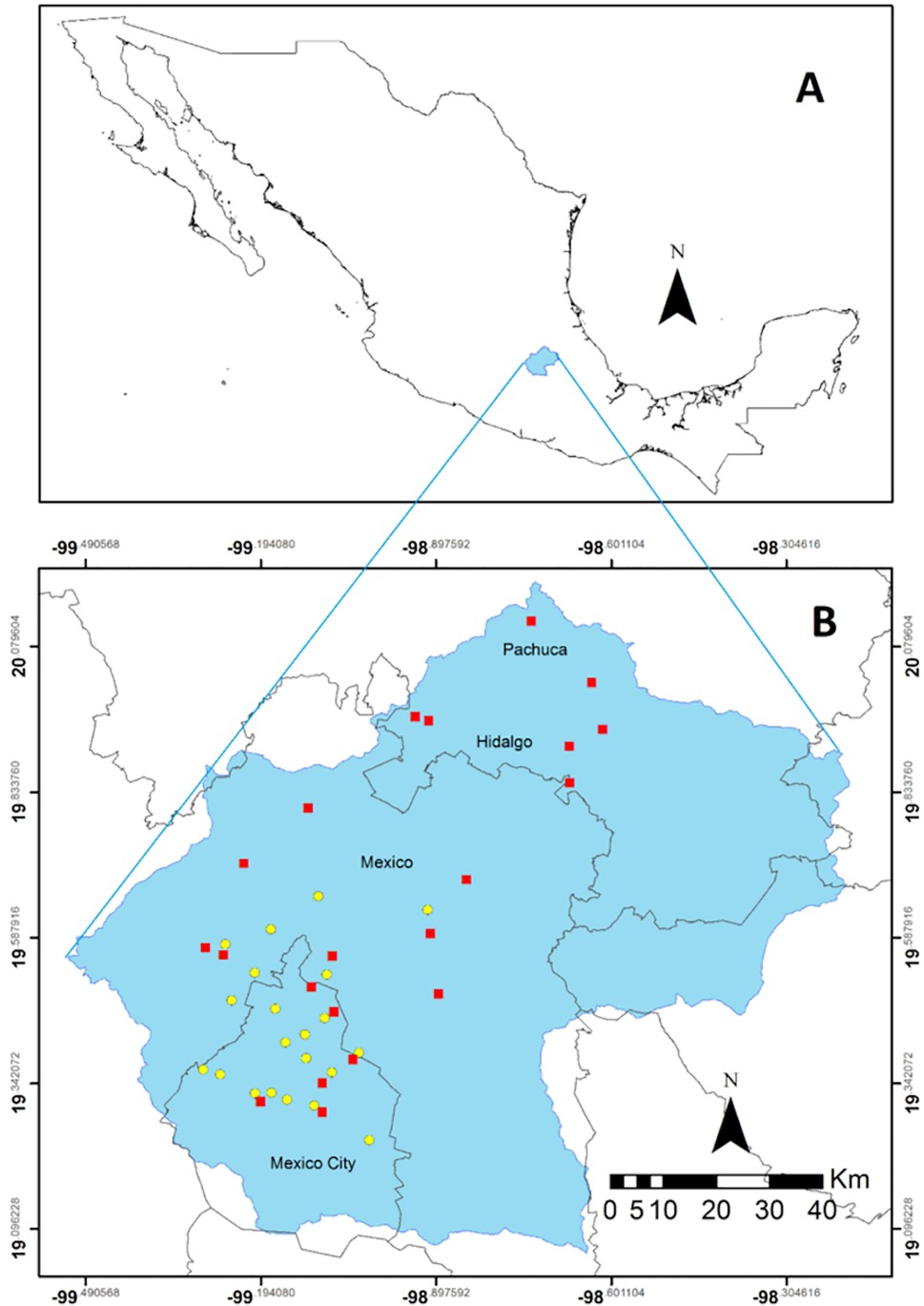

**Figure 1 Region in central Mexico where the study was conducted.** Location of the Valley of Mexico (blue polygon) in Mexico (A). Spatial distribution of the collecting sites throughout the Valley represented by the red squares. The stations belonging to the automatic monitoring network are indicated by yellow dots (B). This network is located between Mexico City and the State of Mexico in the most populated zone of the region.

We determined the relationship between CO concentrations in the Valley of Mexico and the carbon content and the $\delta^{13}C$ values was determined for the epiphytic CAM bromeliad *T. recurvata*, which has an ample distribution in the Americas. This plant has

been utilized as a biomonitor of atmospheric pollution and can be easily found in the Valley of Mexico (*Zambrano et al., 2009*; *Díaz-Álvarez, Lindig-Cisneros & de la Barrera, 2018*; *Díaz-Álvarez & de la Barrera, 2018*; *Piazzetta, Ramsdorf & Maranho, 2018*). Sampling sites were determined in two steps. First, potential sites for the occurrence of *T. recurvata* were identified utilizing Google Earth's satellite scenes, followed by a corroboration by means of the StreetView function where available. In particular, we identified the presence of natural protected areas, vegetation stands in rural/agricultural areas, or parks and other vegetated features in urban areas, where trees and shrubs that could act as phorophytes for *T. recurvata* were be present. A total of 73 sites were identified within the basin. Second, a stratified sampling (within the identified vegetated sites; *Wang et al., 2012*) was conducted on 3–15 November 2014, which occurred towards the end of the rainy season, late in the growing period for *T. recurvata*. After discarding those sites where *T. recurvata* was not found and those where access was not possible, plant samples were collected for elemental and isotopic analyses from 22 sites (Fig. 1), including urban parks (7 sites), built urban structures (4 sites), agricultural sites (6 sites), and natural protected areas (5 sites).

At each site, newly formed, fully developed leaves, which can be visually differentiated from those that grew in previous years, were collected (Permit SGPA/DGGFS/712/2767/14, Secretaría de Medio Ambiente y Recursos Naturales, Mexico) from 5 mature individuals growing at least 5 m apart (*Harmens et al., 2008*; *Díaz-Álvarez et al., 2019*). The samples were dried at 60 °C in a gravity convection oven until reaching constant weight. Sample preparation for stable isotope analyses was conducted following *Díaz-Álvarez & de la Barrera (2018)*. The carbon isotope ratios, reported in parts per thousand were calculated relative to Vienna–Pee Dee Belemnite (VPDB). The analytical precision for the $\delta^{13}$C was 0.2 ± 0.07‰ (SD). The natural abundances of $^{13}$C were calculated as:

$$\delta^{13}C(‰_{\text{versus V–PDB}}) = (R_{\text{sample}}/R_{\text{standard}} \ 1) \times 1000$$

where, $R$ is the ratio of $^{13}$C/$^{12}$C for carbon isotope abundance for a given sample (*Ehleringer & Osmond, 1989*; *Evans, 2001*).

## Statistical and spatial analyses

Data were analyzed following *Díaz-Álvarez & de la Barrera (2018)*. In particular, linear regressions were calculated to determine the relationship between CO concentrations and carbon content (% dry weight), as well as the isotopic composition ($\delta^{13}$C values) for *T. recurvata* in the Valley of Mexico. The differences between sites for the carbon content and the $\delta^{13}$C values were determined by means of the Kruskal–Wallis one-way analysis of variance by ranks, followed by a Nemenyi's post-hoc tests for pairwise multiple comparisons ($p \leq 0.05$). The analyses were conducted using the Pairwise multiple comparison of mean ranks package (PMCMR) in R (version 3.5.3, R Core Team, R foundation for Statistical Computing, Vienna, Austria; *Pohlert, 2014*).

Data interpolation for CO concentration within Mexico City and for *T. recurvata* tissue carbon content and $\delta^{13}$C values throughout the Valley of Mexico were conducted with the

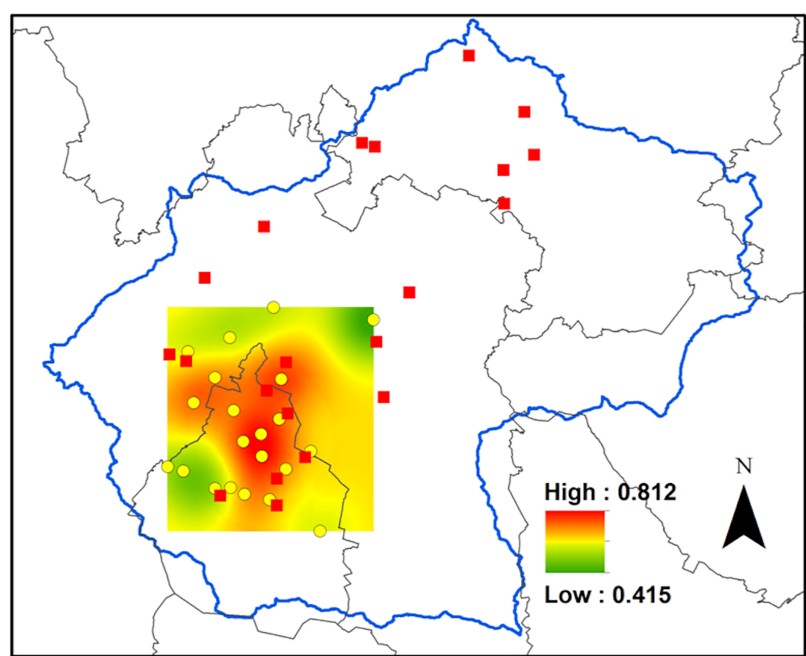

**Figure 2 Carbon monoxide concentrations in Mexico City.** Ordinary kriging for the carbon monoxide concentrations, in parts per million, inside the area covered by the air quality network in the Valley. The data consisted of mean concentration of CO during the period that comprises January to November 2014. The data utilized for this analysis are available at http://www.aire.cdmx.gob.mx.

Ordinary Kriging model contained in ArcGIS 10.5 (ESRI, Redlands, CA, USA), which was also utilized to create the maps. This interpolation method is based on the assumption that data are spatially autocorrelated (*Cressie, 1988*; *Wang et al., 2002*; *Wong, Yuan & Perlin, 2004*), which generally is the case for atmospheric pollutants, as their concentration is higher with proximity to the source, thus influencing the ensuing plant responses (*Stevens et al., 2004*; *Díaz-Álvarez, Lindig-Cisneros & de la Barrera, 2018*). This allows the estimation of parameters of interest in relatively large geographical regions (*Liao, Li & Zhang, 2017*), where only a sparse sampling is available (*Oliver & Webster, 2014*). While interpolation protocols have been developed that improve on Ordinary Kriging, this model is among the most utilized methods in environmental studies, including those considering atmospheric pollution (*Liao, Li & Zhang, 2017*; *Gupta et al., 2018*; *Gómez-Losada et al., 2019*; *Huang et al., 2020*).

# RESULTS

## Spatial distribution of CO concentrations

The CO concentration, averaged 0.67 ± 0.03 ppm in Mexico City, where the monitoring network is deployed (Fig. 2). The central and northern portions had the highest concentrations of CO, reaching a maximum of 0.81 ppm, and the lowest concentration was found at the northeast end of the distribution of the monitoring network, where it reached a mean value of 0.41 ppm (Fig. 2).

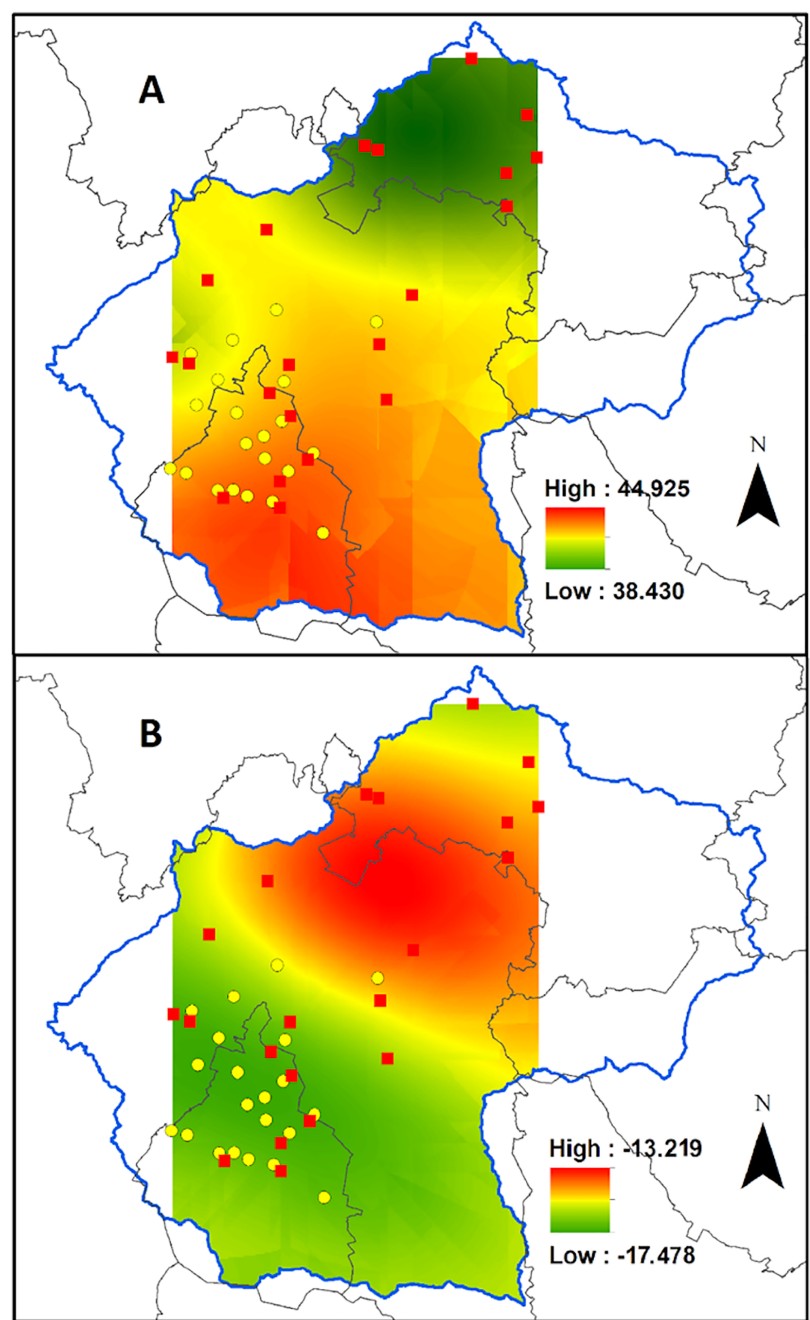

**Figure 3 Biomonitor carbon status in the Valley of Mexico.** Ordinary kriging for the carbon content (A) and δ¹³C values (B) for *Tillandsia recurvata* in the Valley of Mexico.

## Carbon content and isotopic composition for *Tillandsia recurvata*

On average, the tissue carbon content was 42.9 ± 0.3% (dry weight) ranging from 38.4% for plants collected in the urban area of a small town at the north-west portion of the Valley, to 44.9% at the southern part of the Valley, in the middle of the Mexico

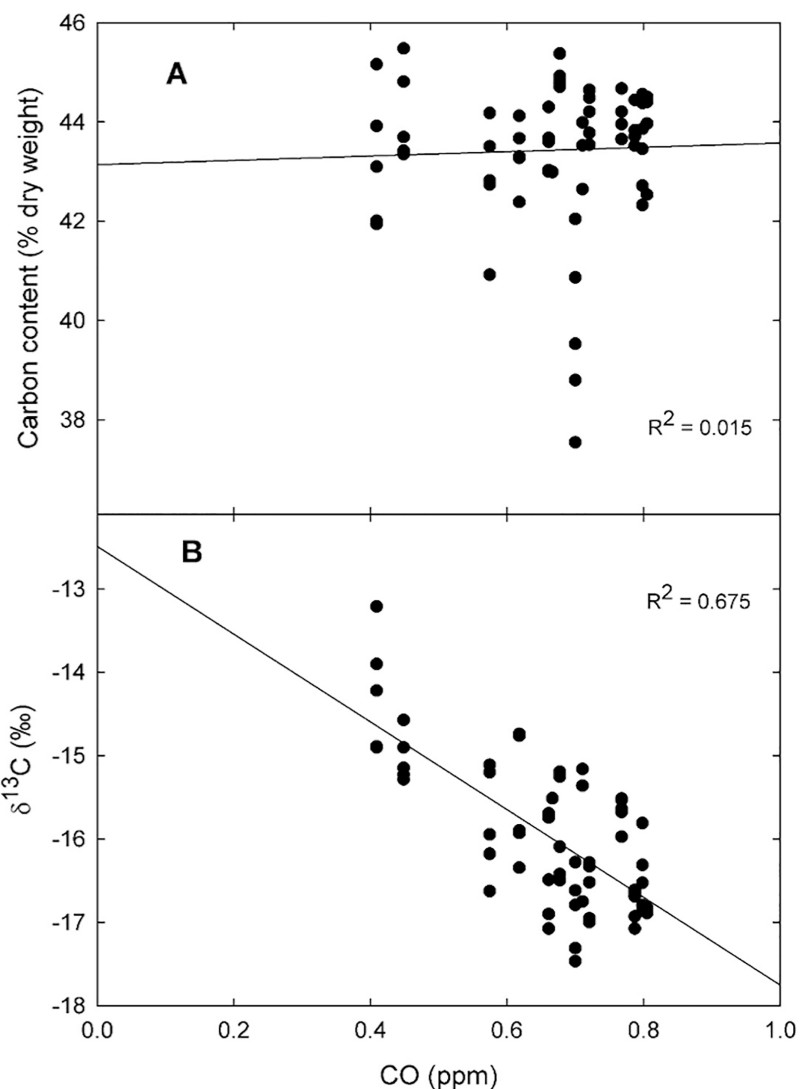

**Figure 4 Biomonitor responses to carbon monoxide.** Linear regression for the relationship between CO concentration in ppm during 2014 and the carbon content (A), and the $\delta^{13}C$ values (B) for *Tillandsia recurvata* at the Valley of Mexico ($n = 5$).

City (Fig. 3A). However, the carbon content was not affected by the CO concentration recorded during 2014 by the Mexico City monitoring network ($R^2 = 0.015$; Fig. 4A).

Contrasting with what occurred for the carbon content, the $\delta^{13}C$ values responded positively to the atmospheric CO concentration, being lower in sites with higher concentrations than in those with lower CO concentrations (Fig. 4B; $R^2 = 0.675$). For example, the $\delta^{13}C$ values reached −16.2‰ in Pachuca, a populous city at the north of the Valley, which is lower than in some rural sites further south (Fig. 3B). The $\delta^{13}C$ values were as negative as −17.5‰ for plants collected in an urban site where the mean CO concentration averaged 0.70 ppm during 2014 (Fig. 3B). In contrast, the highest $\delta^{13}C$ values of −13.2‰ were found for plants growing at the archeological site of Teotihuacan,

whose CO concentration was the lowest recorded inside the monitored area ($p < 0.05$; Fig. 3B).

## DISCUSSION

The atmospheric concentration of CO in the area covered by the monitoring network tended to be higher in the vicinity of the numerous important and busy motorways that have been built in this region and are utilized by the 5.2 million vehicles registered in Mexico City and its metropolitan area, in addition to various thousands of visiting vehicles from other cities and states (*SEDEMA Secretaría del Medio Ambiente de la Ciudad de México, 2016a*). Indeed, motor vehicles are the main source of CO for the region, contributing with 96% of the total emissions, which amount to nearly 700 million tons just in Mexico City; the remaining 4% originated from industrial and domestic sources (*SEDEMA, Secretaría del Medio Ambiente de la Ciudad de México, 2016b*). However, it is worth mentioning that the CO concentrations of 3.8 ppm measured in Mexico City and its metropolitan area during this study did not exceed the Mexican standard (exposure to 11 ppm for 8 h) nor the World Health Organization's (9 ppm) and the United States Environmental Protection Agency's criteria for this pollutant (9 ppm; *World Health Organization WHO, 1999*; *SEMARNAT, 2012*; *SEDEMA Secretaría del Medio Ambiente de la Ciudad de México, 2016a*).

While the carbon content of *T. recurvata* was insensitive to the CO concentration, the $\delta^{13}C$ of newly formed leaves adequately biomonitored the prevailing atmospheric pollution, with values that were substantially more negative in polluted sites than in "clean" sites, a pattern that has been previously documented with atmospheric biomonitors (*Martin, 1994*; *Lichtfouse, Lichtfouse & Jaffrézic, 2003*; *Zambrano et al., 2009*; *Cobley & Pataki, 2019*). When interpreting the isotopic signature of carbon biomonitors it is important to consider that while CO is but a small fraction of the urban carbonaceous emissions, it is produced from combustion simultaneously with $CO_2$, a gas that is usually not included in urban air quality monitoring protocols (*SEMARNAT, 2012*; *SEDEMA, Secretaría del Medio Ambiente de la Ciudad de México, 2016b*). Thus, the plants are in fact mostly recording the isotopic signal of $CO_2$ assimilated by photosynthesis. However, the relationship between atmospheric CO and $CO_2$ concentrations is usually linear, so that the "calibration" conducted in the present study utilizing CO (the only carbon gas that is monitored) as an integrative proxy for urban carbon emissions could be utilized in other unmonitored cities (*Turnbull et al., 2011b*; *Silva, Arellano & Worden, 2013*; *Gromov, Nrenninkmeije & Jöckel, 2017*).

The carbon content in plant tissues is commonly ca. 50% on a dry mass basis, although it varies depending on different factors, such as the developmental stage, organ, species, latitude, water availability, nutrient availability and the $CO_2$ concentration prevalent during organ development (*Díaz-Álvarez, Lindig-Cisneros & de la Barrera, 2015*; *Ma et al., 2018*). High $CO_2$ concentrations, such as those found in urban environments, like Mexico City, are among the factors leading to higher photosynthetic rates in CAM plants (*Drennan & Nobel, 2000*; *Andrade et al., 2007*; *Smith et al., 2009*; *Zotz et al., 2010*; *SEDEMA Secretaría del Medio Ambiente de la Ciudad de México, 2016a*). However, when plants are

exposed to different concentrations of hazardous gases such as NOx, $SO_2$ and CO, the net photosynthetic rate and the photosynthetic pigment content can decrease, because the resulting oxidative stress alters the carboxylation process, leading to a net reduction of the carbon content (*Bytnerowicz et al., 2001*; *Mittler, 2002*; *Muneer et al., 2014*). Such gases are monitored and have actually been detected in areas of Mexico City and Pachuca, although it appears that their concentration is not high enough to cause a tissue carbon reduction for *T. recurvata* (*Díaz-Álvarez & de la Barrera, 2018*; *SEDEMA Secretaría del Medio Ambiente de la Ciudad de México, 2016a*). In this respect, this species displays a physiological resistance to $O_3$ and $SO_2$, which may contribute to the results observed (*Benzing et al., 1992*).

Three factors can help explain the general carbon isotopic pattern observed for *T. recurvata* throughout the study. The first factor is a large difference in the isotopic signature for $CO_2$ from different sources. In particular, the $\delta^{13}C$ values of the air from natural environments can reach –8‰ (*Pichlmayer et al., 1998*; *Widory & Javoy, 2003*). In contrast, the carbon in the air is depleted of $^{13}C$ in sites where motor vehicles and the industrial activities are common. In this regard, the $\delta^{13}C$ values for coal burning, gasoline, diesel and natural gas range from –25 to –42‰ (*Pichlmayer et al., 1998*; *Röckmann et al., 2002*; *Pataki, Bowling & Ehleringer, 2003*; *Widory & Javoy, 2003*; *Semmens et al., 2014*; *Naus, Röckmann & Popa, 2018*).

A second factor is an increasing isotopic discrimination against $^{13}C$ that occurs for CAM plants exposed to high $CO_2$ concentrations (*Zhu, Goldstein & Bartholomew, 1999*). In this case, given that increasing concentrations of $CO_2$ can inhibit the activity of phosphoenolpyruvate carboxylase (PEPc), which is already near saturation at natural concentrations of $CO_2$ (*Ting, 1994*). PEPc has an isotopic discrimination that ranges between 2 and 10‰, which explains the common $\delta^{13}C$ values for CAM plants. However, if PEPc is inhibited, the ribulose bisphosphate carboxylase/oxygenase, an enzyme with a higher discrimination of 22–27‰, will conduct most of the carboxylation, resulting in the observed $\delta^{13}C$ values, which were more negative in polluted sites (*Ehleringer & Osmond, 1989*; *Farquhar, Ehleringer & Hubick, 1989*; *Ting, 1994*; *McNevin et al., 2007*; *Smith et al., 2009*; *Cernusak et al., 2013*).

A third potential factor influencing the observed pattern is air temperature in cities. Warmer than natural nocturnal air temperatures, such as those resulting from the urban heat island in Mexico city, drive changes in the carbon fixation cycle of CAM plants, which in turn leads to an increased isotopic discrimination against $^{13}C$ (*Troughton & Card, 1975*; *Farquhar, Ehleringer & Hubick, 1989*; *Jauregui, 1997*; *Zhu, Goldstein & Bartholomew, 1999*; *Cui & De Foy, 2012*; *Cernusak et al., 2013*).

## CONCLUSIONS

Owing to the close relationship observed between the isotopic composition of *T. recurvata* and the atmospheric CO concentration in the Valley of Mexico, the $\delta^{13}C$ values can be useful for characterizing carbonaceous pollution. In addition, given that the emissions of CO and $CO_2$ are accompanied by the emission of other pollutants, such as NOx, $SO^2$, and

heavy metals, which results in the formation of secondary pollutants such as $O_3$ and particulate matter, this plant can be deemed as an ideal candidate for implementing broader monitoring studies in regions where automatic monitoring networks are not available and the bromeliad is abundant.

## ACKNOWLEDGEMENTS

We thank Mr. F.S. Zúñiga for his hospitality in Mexico City and the generosity of the personnel of the Zona Arqueológica de Teotihuacán, especially Dr. V. Ortega.

### Funding

This work was supported by the Dirección General de Asuntos del Personal Académico, Universidad Nacional Autónoma de México (Grants PAPIIT IN205616 and IN211519). The funders had no role in study design, data collection and analysis, decision to publish, or preparation of the manuscript.

### Grant Disclosures

The following grant information was disclosed by the authors:
Dirección General de Asuntos del Personal Académico, Universidad Nacional Autónoma de México: PAPIIT IN205616 and IN211519.

### Competing Interests

The authors declare that they have no competing interests.

### Author Contributions

- Edison A. Díaz-Álvarez conceived and designed the experiments, performed the experiments, analyzed the data, prepared figures and/or tables, authored or reviewed drafts of the paper, and approved the final draft.
- Erick de la Barrera conceived and designed the experiments, performed the experiments, authored or reviewed drafts of the paper, and approved the final draft.

### Field Study Permissions

The following information was supplied relating to field study approvals (i.e., approving body and any reference numbers):
  Field studies were conducted under collection permit by the Secretaría de Medio Ambiente y Recursos Naturales (SGPA/DGGFS/712/2767/14).

### Data Availability

  Raw data are available as a Supplemental File.

### Supplemental Information

Supplemental information for this article can be found online at http://dx.doi.org/10.7717/peerj.9283#supplemental-information.

![PeerJ]

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
