# Peer review of "Isotopic biomonitoring of anthropic carbon emissions in a megalopolis"

_PeerJ, doi:10.7717/peerj.9283_

## Round 0.1 · original submission · Major Revisions

I have received three reviews for this manuscript. The reviewers comments make it clear that the manuscript will need significant work before it is suitable for publication in PeerJ. Please read the reviewers comments carefully and respond to each of them in turn. In particular, the reviewers have expressed significant concerns around the methodology and the statistical analysis. Consequently, it is likely that the Results and Discussion section will need to be rewritten.

·

Basic reporting

The article is written in clear and professional English. I believe that literature is sufficient to contextualize, analyze and discuss the results obtained. The structure of the article, as well as the quality of the figures is adequate.

Experimental design

I think the work is consistent with the objectives of the journal. The research questions are well defined and relevant since they propose an alternative of air quality monitoring using plants, which implies low cost of analysis. The investigation was conducted rigorously and the technical standards adequate and sufficient to lower the stated objectives. It is necessary to make some clarifications to the methodology to be completely clear, as noted below.

Validity of the findings

The article presents and validates innovative results in terms of air quality monitoring strategies. The statistical analyzes performed are adequate to support the conclusions raised by the authors, which are completely linked to the objectives, results and analyzes developed.

Additional comments

I suggest that authors comment the importance to monitor CO in particular if it represents only a fraction of the pollutants emitted into the atmosphere as the authors themselves point out.

Although the methodology used is adequate to meet the objectives set, the authors comment that they made the collection of plant material during the growth period, however it is necessary to specify that they worked with new leaves, as well as give an idea of the foliar growth rate in the species used. This is important given that the technique used for the quantification of 13C characterizes the isotopic proportion fixed during leaf formation, so it must be ensured that the growth of the leaf used was during the same period in which they average the CO concentration of their work sites. In this way, it is ensured that the calculated correlation is adequate.

Line 92. Please specify that is above sea level.
Line 93. Please specify that is elevation above sea level.

Reviewer 2 ·

Basic reporting

This paper illustrates the biomonitoring of CO emissions using δ13C values in tillandsia. In my opinion, the article has too many significant issues, that can not be easily fixed, even with an extensive revision of the manuscript.
The abstract sounds a bit confused: for example, there is no need to specify here some statistical details about the significance of the tests or other methodological parameters. On the other hand, the aim of the paper must be stated much more clearly than it is now.
I am puzzled about the choice of the pollution marker made in this paper. Although fuel combustion indeed generates CO and CO2, but also many other gaseous and non-gaseous compounds and/or elements that can be much more easily linked to the emission. The authors report some of them in the methods (e.g. line 103: O3, NOx, NO2, NO, PM10 PM2.5).
As the authors state, the amounts of emitted CO from combustion is "very low or insignificant" if compared to CO2 emission. That makes the use of this marker quite problematic.
The aim of the paper is not clearly stated. Also, some crucial aspects of the method used are not enough introduced. A general audience may not be familiar with the relationship occurring between CO emissions and δ13C values accumulated in the plant.
There are also some conceptual misunderstanding of biomonitoring techniques (e.g. line 71: in fact, they are a complementary approach to air quality monitoring and do not substitute chemical-physical procedures. They give additional, biological information on the response to the pollution that can not be retrieved from chemical data by themselves.
I also see major problems in the sampling design. I acknowledge that the distribution of tillandsia is not regular. Still, there are dozen of sampling designs that can face this issue and save the probabilistic base of the location of sampling points. Even though a preferential sampling may be an option in some cases, it generates critic issues when data are used for making inference on non-sampled locations in the study area (e.g. mapping through interpolation), as in the case of this paper.
These problems become more and more critical when trying to put in correlation δ13C values and interpolated CO values with an unknown error.
Moreover, the discussion is in some parts too much speculative and not always related to the results presented in the study (e.g. 222-227).

Experimental design

Major issues both in the sampling design and in the definition of the statistical model.

Validity of the findings

Although there is some useful perspective, the issues described above make it difficult to come to robust conclusions.

·

Basic reporting

The manuscript evaluate changes in “Tissue carbon content and the δ13C values for the epiphytic CAM bromeliad Tillandsia recurvata and the relationship of both parameters with the CO concentrations in the Valley of Mexico basin”. This research is novel and the topic fits well to the scope of the journal, because it is the first studie to investigating relationship between carbon emissions and the plant´s isotopic composition. However, my main concern about this manuscript is statistycal analysis related with desing. I think that land use types (urban parks, agricultural and natural protected areas), should be included as explanatory factor in regression models. Thus, I suggest to explicate how many sites (22 sites in total) were distributed within of urban parks, agricultural and natural protected areas in this study??. In addition, I suggest to include CO concetrations and land use types as explicator factors in the new multiple regresion model (normal distribution) or Generalized linear model (not normal distribution).

References

Please the authors should include all references following authors guide of journal

Correct format

Sala OE, Chapin III SF, Armesto JJ, Berlow E, Bloomfield J, Dirzo R, Huber-Sanwald E, Huenneke LF, Jackson RB, Kinzig A, Leemans R, Lodge DM, Mooney HA, Oesterheld M, Poff NL, Sykes MT, Walker BH, Walker M, Wal DH (2000) Global biodiversity scenarios for the year 2100. Science 344 287:1770–1774. doi: 10.1126/science.287.5459.1770.

Incorrrect format

Silva, S. J., Arellano, A. F., & Worden, H. M. (2013). Toward anthropogenic combustion emission constraints from space-based analysis of urban CO2 /CO sensitivity. Geophysical Research 349 Letters, 40(18), 4971–4976. doi:10.1002/grl.50954

Experimental design

The manuscript evaluate changes in “Tissue carbon content and the δ13C values for the epiphytic CAM bromeliad Tillandsia recurvata and the relationship of both parameters with the CO concentrations in the Valley of Mexico basin”. This research is novel and the topic fits well to the scope of the journal, because it is the first studie to investigating relationship between carbon emissions and the plant´s isotopic composition. However, my main concern about this manuscript is statistycal analysis related with desing. I think that land use types (urban parks, agricultural and natural protected areas), should be included as explanatory factor in regression models. Thus, I suggest to explicate how many sites (22 sites in total) were distributed within of urban parks, agricultural and natural protected areas in this study??. In addition, I suggest to include CO concetrations and land use types as explicator factors in the new multiple regresion model (normal distribution) or Generalized linear model (not normal distribution).

2. Materials and Methods
Line 90: Study region
I suggest to explicate how many sites (22 sites in total) were distributed within of urban parks, agricultural and natural protected areas in this study.
Line 100 and 109: Atmospheric CO concentration and Biomonitoring

I suggest to join the following subtitles, for instance:

Atmospheric CO concentration and Biomonitoring
Geoestatitical and Statistical analysis

Line 131-138: Linear regressions were calculated to determine the relationship between CO concentrations and carbon content (% dry weight), as well as the isotopic composition (δ13C values) for Tillandsia recurvata in the Valley of Mexico. The differences between sites for the carbon content and the δ13C values were determined by means of the Kruskal-Wallis one-way analysis of variance by ranks, followed by a Nemenyi´s post-hoc tests for pairwise multiple comparisons (p ≤ 0.05). The analyses were conducted using the Pairwise multiple comparison of mean ranks package (PMCMR) in R (version 3.5.3, R Core Team, R foundation for Statistical Computing, Vienna, Austria; Pohlert, 2014).

In my opinion, the tissue carbon content and the δ13C values for the epiphytic CAM bromeliad Tillandsia recurvata and the relationship of both parameters with the CO concentrations and land use types can be analyzed by multiple regresion model (normal distribution) or Generalized linear model (not normal distribution), and post-hoc Tukey test.


3. Results and Discsussion.

In my opinion,these sections should be rewritten with results the new statistycal analysis.

Validity of the findings

2. Materials and Methods
Line 90: Study region
I suggest to explicate how many sites (22 sites in total) were distributed within of urban parks, agricultural and natural protected areas in this study.
Line 100 and 109: Atmospheric CO concentration and Biomonitoring

I suggest to join the following subtitles, for instance:

Atmospheric CO concentration and Biomonitoring
Geoestatitical and Statistical analysis

Line 131-138: Linear regressions were calculated to determine the relationship between CO concentrations and carbon content (% dry weight), as well as the isotopic composition (δ13C values) for Tillandsia recurvata in the Valley of Mexico. The differences between sites for the carbon content and the δ13C values were determined by means of the Kruskal-Wallis one-way analysis of variance by ranks, followed by a Nemenyi´s post-hoc tests for pairwise multiple comparisons (p ≤ 0.05). The analyses were conducted using the Pairwise multiple comparison of mean ranks package (PMCMR) in R (version 3.5.3, R Core Team, R foundation for Statistical Computing, Vienna, Austria; Pohlert, 2014).

In my opinion, the tissue carbon content and the δ13C values for the epiphytic CAM bromeliad Tillandsia recurvata and the relationship of both parameters with the CO concentrations and land use types can be analyzed by multiple regresion model (normal distribution) or Generalized linear model (not normal distribution), and post-hoc Tukey test.


3. Results and Discsussion.

In my opinion,these sections should be rewritten with results the new statistycal analysis.

4. Conslusions

Line 334: pollutants such as NOx, SO2, O3, heavy metals, particulate matter, etc

In my opinion, authors should deleted the follow word “etc.”

Additional comments

PeerJ-41903

Dear Editor,

The manuscript evaluate changes in “Tissue carbon content and the δ13C values for the epiphytic CAM bromeliad Tillandsia recurvata and the relationship of both parameters with the CO concentrations in the Valley of Mexico basin”. This research is novel and the topic fits well to the scope of the journal, because it is the first studie to investigating relationship between carbon emissions and the plant´s isotopic composition. However, my main concern about this manuscript is statistycal analysis related with desing. I think that land use types (urban parks, agricultural and natural protected areas), should be included as explanatory factor in regression models. Thus, I suggest to explicate how many sites (22 sites in total) were distributed within of urban parks, agricultural and natural protected areas in this study??. In addition, I suggest to include CO concetrations and land use types as explicator factors in the new multiple regresion model (normal distribution) or Generalized linear model (not normal distribution).
* * *
Further comments:
1. Introduction
Line 37: Atmospheric pollution has become a serious threat for human health and the environment.

Here, authors should add reference
Line 39-41: For example, Mexico City, with vertiginous growth of industrial and household activities, as well as numerous motor vehicles of all classes, has seen an increase of emissions of different pollutants to the atmosphere.
Here, authors should add reference

Line 47-51: The combustion of carbonaceous fuels generates two main carbon products, carbon monoxide and carbon dioxide. However, the proportion in which they are emitted can vary. For example, when a motor works in optimal conditions, that is when the mixture of air, fuel, and temperature inside an automobile engine is ideal, the complete combustion of the fuel is achieved (high efficiency), resulting in the complete oxidation of carbon and subsequent emission of CO2.

Here, authors should add reference

Line 80-81: Although, the carbon isotopic composition of this plant has been
reported for polluted and non-polluted sites

Here, authors should add reference

2. Materials and Methods
Line 90: Study region
I suggest to explicate how many sites (22 sites in total) were distributed within of urban parks, agricultural and natural protected areas in this study.
Line 100 and 109: Atmospheric CO concentration and Biomonitoring

I suggest to join the following subtitles, for instance:

Atmospheric CO concentration and Biomonitoring
Geoestatitical and Statistical analysis

Line 131-138: Linear regressions were calculated to determine the relationship between CO concentrations and carbon content (% dry weight), as well as the isotopic composition (δ13C values) for Tillandsia recurvata in the Valley of Mexico. The differences between sites for the carbon content and the δ13C values were determined by means of the Kruskal-Wallis one-way analysis of variance by ranks, followed by a Nemenyi´s post-hoc tests for pairwise multiple comparisons (p ≤ 0.05). The analyses were conducted using the Pairwise multiple comparison of mean ranks package (PMCMR) in R (version 3.5.3, R Core Team, R foundation for Statistical Computing, Vienna, Austria; Pohlert, 2014).

In my opinion, the tissue carbon content and the δ13C values for the epiphytic CAM bromeliad Tillandsia recurvata and the relationship of both parameters with the CO concentrations and land use types can be analyzed by multiple regresion model (normal distribution) or Generalized linear model (not normal distribution), and post-hoc Tukey test.


3. Results and Discsussion.

In my opinion,these sections should be rewritten with results the new statistycal analysis.

4. Conslusions

Line 334: pollutants such as NOx, SO2, O3, heavy metals, particulate matter, etc

In my opinion, authors should deleted the follow word “etc.”

5. References

Please the authors should include all references following authors guide of journal

Correct format

Sala OE, Chapin III SF, Armesto JJ, Berlow E, Bloomfield J, Dirzo R, Huber-Sanwald E, Huenneke LF, Jackson RB, Kinzig A, Leemans R, Lodge DM, Mooney HA, Oesterheld M, Poff NL, Sykes MT, Walker BH, Walker M, Wal DH (2000) Global biodiversity scenarios for the year 2100. Science 344 287:1770–1774. doi: 10.1126/science.287.5459.1770.

Incorrrect format

Silva, S. J., Arellano, A. F., & Worden, H. M. (2013). Toward anthropogenic combustion emission constraints from space-based analysis of urban CO2 /CO sensitivity. Geophysical Research 349 Letters, 40(18), 4971–4976. doi:10.1002/grl.50954



Sincerely,

PhD. Angel Raimundo Benitez Chavez.
Departamento de Ciencias Biológicas.
Sección de Ecología y Sistemática
Herbario HUTPL
Universidad Técnica Particular de Loja
Telf: 072570275 Ext: 3034, Celular: 0985182195
San Cayetano Alto. Campus Universitario. Loja-Ecuador

---

## Round 0.2 · Major Revisions

You will see that although reviewer 2 notes improvements in the manuscript, there are still concerns. Please could you act on the comments which are intended to be constructive to help improve the manuscript. In particualr, being clearer about the role of biomonitoring, expanding on the methods as requested rather than just referring to other papers and adding more on the applicability of your results.

Reviewer 2 ·

Basic reporting

I must say that the objectives of the article are much clearer in the rebuttal letter than in the first version of the manuscript. In this regard, the revisions in the second version sufficiently compensate for the limits previously found.

I have the impression that in some cases the authors are not very likely to evaluate the reviewers' suggestions. Some examples:

I have no doubt that there is no misunderstanding on their part about the role of biomonitoring. What I mean is that if they write "in lieu of" (L75) an unskilled reader could understand that biomonitoring can replace a chemical-physical approach. And this is not true.

The authors say that “We have to stand by our sampling design and data analyses”.
I can fully understand that the authors cannot think of further sampling and must manage the dataset available to them. However, this does not necessarily make their sampling adequate or robust. I do not agree very much in recalling all the details of the sample design to a work published elsewhere. I would suggest adding at least a few more details.

Looking at the results of the interpolation map in the south-eastern part of their study area, it seems clear to me that there are some issues related to the uneven distribution of the sample points with respect to the area on which the model was run. See for example the net response patterns in Figure 3A. I repeat to avoid misunderstandings: I am more than aware that there are differences between perfect sampling plan and plans applicable in the real world. However, I would suggest to the authors, for example, to insert some information on the variability of the geostatistical model to allow the reader to acquire more information on the areas of applicability of their method.

Experimental design

see basic reporting

Validity of the findings

see basic reporting

Additional comments

see basic reporting

·

Basic reporting

PeerJ-41903

Dear Editor,

The authors has taken into consideration changed in the text, which has resulted in a considerable improvement of introduction, methods and disccusion of the manuscript entitled "Mapping pollution in a megalopolis: Isotopic biomonitoring of anthropic carbon emissions". In addition, the total samples number in each site (122 samples) is addecuate for data analysis. Thus, this research is novel and the topic fits well to the scope of the journal, because it is the first studie to investigating this topic.


Sincerely,

PhD. Angel Raimundo Benitez Chavez.
Departamento de Ciencias Biológicas.
Sección de Ecología y Sistemática
Herbario HUTPL
Universidad Técnica Particular de Loja
Telf: 072570275 Ext: 3034, Celular: 0985182195

Experimental design

The authors has taken into consideration changed in the text, which has resulted in a considerable improvement of introduction, methods and disccusion of the manuscript entitled "Mapping pollution in a megalopolis: Isotopic biomonitoring of anthropic carbon emissions". In addition, the total samples number in each site (122 samples) is addecuate for data analysis. Thus, this research is novel and the topic fits well to the scope of the journal, because it is the first studie to investigating this topic.

Validity of the findings

The authors has taken into consideration changed in the text, which has resulted in a considerable improvement of introduction, methods and disccusion of the manuscript entitled "Mapping pollution in a megalopolis: Isotopic biomonitoring of anthropic carbon emissions". In addition, the total samples number in each site (122 samples) is addecuate for data analysis. Thus, this research is novel and the topic fits well to the scope of the journal, because it is the first studie to investigating this topic.

Additional comments

The authors has taken into consideration changed in the text, which has resulted in a considerable improvement of introduction, methods and disccusion of the manuscript entitled "Mapping pollution in a megalopolis: Isotopic biomonitoring of anthropic carbon emissions". In addition, the total samples number in each site (122 samples) is addecuate for data analysis. Thus, this research is novel and the topic fits well to the scope of the journal, because it is the first studie to investigating this topic.

---

## Round 0.3 · accepted · Accept

Thank you for making the changes suggested from the second review. I am now happy to accept this manuscript for publication in PeerJ.